# Improving the Treatment Outcome of Naso-Orbito-Ethmoido-Maxillary Fractures Using Virtual Three-Dimensional Anthropometric Data

**DOI:** 10.3390/ijerph191610412

**Published:** 2022-08-21

**Authors:** Andrei-Mihail Roșu, Daniela Șulea, Geanina Bandol, Bogdan Mihail Cobzeanu, Liliana Moisii, Florentina Severin, Luiza-Maria Cobzeanu, Dragoș Negru, Oana Cristina Roșu, Dragoș Octavian Palade, Victor Vlad Costan, Mihail Dan Cobzeanu

**Affiliations:** 1Surgical Department, Faculty of Medicine, University of Medicine and Pharmacy “Grigore T. Popa”, 700115 Iași, Romania; 2Emergency Clinical Hospital “Sfântul Spiridon” Iași, 700111 Iași, Romania; 3Surgical Department, Faculty of Dental Medicine, University of Medicine and Pharmacy “Grigore T. Popa”, 700115 Iași, Romania; 4Regional Institute of Oncology, 700483 Iași, Romania; 5Clinical Rehabilitation Hospital, 700661 Iași, Romania; 6Department of Pneumology, Emergency Clinical County Hospital, 730006 Vaslui, Romania

**Keywords:** NOEM fracture, trauma, midface proportions, 3D

## Abstract

Background: Naso-orbito-ethmoido-maxillary (NOEM) fractures are usually the result of a high or moderate intensity impact to the upper midface. These types of fractures are difficult to treat and are frequently misdiagnosed. Craniometric analysis can be of real aid in the treatment of NOEM complex fractures by establishing midfacial proportions. Aim: This study aims to establish the distances between selected anthropometric points and midfacial proportions found in the adult Caucasian population and to determine if any differences exist between genders. Methodology: Measurements between anthropometric points, nasion (N), dacryon (D), infraorbital foramen (IOF), frontomalare orbitale (FMO), rhinion (Rhi) and porion (Po), were made on 3D models obtained using patients’ CT exams. Results: Significant differences were found between genders for the orbital dimensions represented by N–FMO (*p =* 0.000), N–IOF (*p =* 0.000), Rhi–FMO (*p* = 0.000), Rhi–IOF (*p =* 0.000), nose bridge width N–D (*p =* 0.001), Rhi–D (*p =* 0.016), D–D (*p =* 0.038) and the projection of the nose evaluated by Rhi–Po (*p =* 0.000), N–Po (*p =* 0.000), while a *t*-test showed that there are no significant differences between males and females for the N–Rhi (*p* = 0.254). Conclusions: The values of these measurements can be utilized during skeletal reconstruction after NOEM fractures, especially for bilateral comminuted fractures where no points of comparison are available.

## 1. Introduction

The naso-orbito-ethmoido-maxillary complex (NOEM) is represented by the confluence of the frontal sinuses within the frontal bone, the ethmoid bone containing the ethmoidal sinuses, the maxillary bone, the medial orbital wall and the nasal bones. Due to the intricate anatomical relationships, fractures at this level are difficult to diagnose, sometimes being confused with nasal pyramid fractures, which delays the optimal treatment time.

NOEM complex fractures are caused by forces applied to the midface. Due to the violent impact involved in producing this type of fractures, they are accompanied by other injuries of the face or other parts of the body. Road accidents, especially those involving an occupant who was not wearing a seat belt at the time of impact, are the most common causes of trauma affecting the NOEM complex. These represent 4% of all skull fractures in adults [1,2], the ones most affected being young adult males.

Craniometric analysis is the main method used to determine the proportions of a skull, allowing for cranial structures identification and characterization of bony features in different populations [3]. Until a few years ago, the main method to obtain these measurements and proportions was done manually, by direct measurements on human skulls [4].

With the advancement of medical imaging and 3D modelling, a new opportunity arises, where the craniometric analysis can be done virtually, on accurate models of human skulls.

Knowing these values in each population is a great aid in forensic medicine or in reconstructive surgery, particularly where, due to the severity of the defect and comminution, certain bony structures have to be reconstructed with the aid of bone grafts or titanium implants.

Due to the increasing number of severe midfacial and NOEM complex trauma cases observed during clinical practice, we consider it necessary to establish the mean distances between different craniometric points and to assess if any differences exist between genders, measured on three-dimensional (3D) models of Caucasian adult patients in Romania, although these values can be used on all individuals with the same racial and ethnic characteristics. These measurements could be useful both for the diagnosis and for the personalized treatment of NOEM complex fractures.

## 2. Materials and Methods

Between 2018 and 2021, a study was conducted in the otorhinolaryngology and oro-maxillofacial surgery clinics within the emergency clinical hospital “Sfântul Spiridon” from Iași, the largest emergency hospital serving the eastern and north-eastern part of Romania. The study was performed with the approval of the ethics committee of the University of Medicine and Pharmacy “Grigore T. Popa” Iași, taking into account the ethical aspects and the protection of personal data. The purpose of the study was to establish the distances between different craniometric points measured on virtual 3D models obtained from the anonymized computer tomography (CT) examinations of 113 hospitalized patients, aged between 19 and 93 years (Figure 1). The patients of both sexes had no history of traumatic injuries, tumours, malformations, or surgery on the midface that could have interfered with the performed measurements by possible differences in the position of the anthropometric points. The lot was divided according to gender, with 55 females and 45 males included in the study. The exclusion criteria were represented by cases in which the CT volumes did not include the entire midface (5 patients), cases in which there were scanning artefacts overlapping the areas of interest, or cases in which the landmarks for the anthropometric point placement were not clear (8 patients).

The following anthropometric landmarks were selected:Nasion (N): the most anterior point located at the middle line where the frontal and nasal bones meet [5];Dacryon (D): the point of junction of the maxillary, lacrimal and frontal bones [5];Infraorbital foramen (IOF): the anterior opening of the infraorbital canal;Rhinion (Rhi): the anterior tip of the nasal bones suture [5];Frontomalare Orbitale (FMO): the fronto-malar suture [6];Porion (Po): The most superior point on the upper margin of the external auditory meatus [6].

The anthropometric points were marked on the CT slices using the 3D MPR (multiplanar reconstruction) view (Figure 2) according to the definition of each point. The 3D models were obtained by transforming Digital Imaging and Communications in Medicine (DICOM) data into stereolithographic models using specific CT viewing software (OsiriX MD; Pixmeo SARL—266 Rue de Bernex—CH1233 Bernex—Switzerland) by using the 3D surface rendering function with a predefined pixel value of 300 (Figure 3) and then exporting it as an .OBJ file. The DICOM data were obtained using the Siemens Somatom Emotion 16 CT scanner (Siemens Healthcare GmbH Henkestr. 127, 91052 Erlangen, Germany). The acquisition was performed in the axial plane with slices of 1 mm thickness. The actual measurements (Figure 4A,B) of N–FMO, N–Rhi, N–IOF, N–D, Rhi–FMO, Rhi–D, Rhi–IOF, Rhi–Po, N–Po and D–D distances were performed using the Meshlab program (Visual ComputingLab, Thermi Rd, 57001, Thessaloniki, Greece).

The marking of the anthropometric points, measurements and verification were carried out by 3 maxillofacial surgeons with 9, 12 and over 30-years work experience. If any landmark was not clearly visible, the 3D model was discarded.

The statistical analysis was performed using IBM SPSS Statistics 26 (IBM Corp. Released 2019, IBM SPSS Statistics for Windows, Version 26.0, Armonk, NY, USA). To evaluate the data distribution, the Kolmogorov–Smirnov and Shapiro–Wilk tests were used. The results were compared using the *t*-test and were considered significant at a “*p*” value lower than 0.05 (*p* < 0.05).

## 3. Results

Our study shows significant differences between females and males for the distances regarding the orbit dimensions N–FMO (*p =* 0.000), N–IOF (*p =* 0.000), Rhi–FMO (*p =* 0.000), Rhi–IOF (*p =* 0.000), nose bridge width N–D (*p =* 0.001), Rhi–D (*p =* 0.016), D–D (*p =* 0.038) and for the projection of the nose Rhi–Po (*p =* 0.000), N–Po (*p =* 0.000) while no significant differences were found between males and females for N–Rhi (*p =* 0.254) (Figure 5, Table 1).

Statistical analysis using *t*-test were performed for age groups (18–60 years, respectively, ≥61 years). The p value was considered significant at a 0.01 level, showed no significant differences when comparing the measured distances between age groups, with the lowest *p* value of 0.062 obtained for the N-D measures.

The distances between the craniometric points are presented in Table 1 with the minimum, maximum and mean values for each distance and for each gender.

Both the Kolmogorov–Smirnov test, as well as the Shapiro–Wilk test results suggest that variables follow a normal distribution in the entire population.

## 4. Discussion

NOEM complex fractures involve the ethmoid sinuses, orbit, nose and the medial maxillary buttress. They are usually the result of moderate-to-high intensity impact to the upper 1/3 of the midface. In other cases, they are the result of an isolated impact on the nasal root [7], mainly caused by road accidents, domestic violence, sports injuries and industrial trauma [8]. These lesions may occur in the case of fractures of the midface, may be isolated or bilateral and may have different fracture patterns on each side of the midline, depending on the mechanism and speed of impact [9,10].

Severe fractures with significant displacement of bone fragments often require an open reduction and internal fixation (ORIF). The type and classification of the lesion dictate the approach. Therefore, type I Markovitz fractures can be managed by a trans-vestibular approach, combined with a trans-orbital approach [11], while Markovitz type II and type III fractures require either an open sky approach, a coronal incision or midfacial degloving. After reduction and fixation, medial cantal tendon attachment should be evaluated and medial canthopexy should be performed if necessary.

The most frequent complications connected with NOEM fractures (telecantus, facial deformity, diplopia, epiphora, enophthalmos and midface retrusion) [12,13] result from an inadequate treatment or from the missed optimal repair time due to misdiagnosis, the condition of the patient, or rejection of general anaesthesia. The amendment of these sequelae requires surgery or a corrective action that can be achieved by static reanimation with barbed wires, a procedure that is more suitable for patients who reject or are not fit for general anaesthesia [14].

With the results showing significant differences between males and females for the N–D, Rhi–D, D–D, N–FMO and Rhi–FMO distances, we can conclude that the width of the orbit, as well as the width of the bridge of the nose differs between the two sexes, with the same being said regarding the projection of the nose evaluated by N–Po and Rhi–Po distances. However, there seem to be no gender-related differences regarding the length of the nose since there were no significant differences between Rhi–N distances in males and females.

The D–D distance, corresponding to the intercanthal distance, represents an area of maximum interest considering, primarily, the aesthetic consequences. Even the smallest changes and asymmetries appear unsightly, in addition to the functional problems they cause. There are studies that describe the importance of this intercanthal distance in terms of aesthetics, with a number of features, concerning the perception of the beauty and personality of a person [15].

The average distance between the inner corners of the eyes is between 29 and 34 mm in adult female patients and between 29 and 36 mm in male patients [10]. The intercanthal distance is different from the measurements between the bilateral D-points from our study because of the thickness of the soft tissues in the region, especially the medial cantal tendon (MCT). MCT consists of two layers, shallow and deep, 9.6 mm long, 2.4 mm wide and 4.5 mm thick for the surface layer, respectively, 3.7 mm long, 2.9 mm wide and 0.3 mm thick for the deep layer [16].

In a study published in 2022, comparing 3D measured craniometric parameters in patients with and without endocrine orbitopathy, the results showed a mean value for the anterior inter orbital width (D–D) of 21.3 ± 2.7 mm. For the male healthy participants, the distance was 21.1 ± 2.7 mm, and for the female subjects in the same non endocrine orbitopathy group was 21.6 ± 2.6 mm [17]. Compared to our findings, we observe a small difference of less than 2 mm.

A study evaluating the accuracy of 3D skulls manufactured by fused deposit modelling found that the distance between nasion and the medial orbit, corresponding with the D point, was 16.85 mm and the N–Po distance had a value of 109.22 mm [18]. Although we obtained a slightly smaller value for N–D, the result was similar regarding the N–Po distance for both males and females.

In our study, we found that there are no statistically significant differences between genders regarding the nose length. This is in accordance with other studies [19,20].

Przygocka et al. [21] acquired from the analysis made on human skulls of Polish adults, the typical N–IOF distance of 45.00 mm, 39.00 mm for Rhi–IOF and 19.00 mm for N–Rhi. Compared to our results, we obtained approximately same values for N–IOF and Rhi–IOF, but the N–Rhi value was 3–4 mm smaller.

Although minor, some of the differences in the values found in our study and those presented in the literature could be due to the selection of a different population group.

There are studies that underline the importance of surgeons knowing the anthropometric proportions of the face, particularly in fields dealing with facial reconstruction and aesthetic changes, in order to preserve the facial features as close as possible to the morphology of that population [22].

The use of three-dimensional printed stereolithographic models for shaping implantable material reduces operating time and, at the same time, ensures the accuracy of facial reconstruction. This procedure, called EPPOCRATIS by some authors [23], has been used successful in the case of trauma to the midface.

Despite the fact that FMO and IOF craniometric landmarks are not considered by all clinicians [24,25], they were used on the grounds of being somewhat steady during human existence and effectively perceived on standard antero-posterior radiographs of the skull. The obtained outcome agree with the typical findings by different authors who estimated the distance between these points.

The geography of the infraorbital foramen (IOF) represents a significant marker in facial medical procedures [1,2,25], and it may very well be utilized as a source of perspective for additional reestablishment if an occurrence of injury should arise.

Another utility for these measurements is found in cases with severe trauma and comminuted fractures of the entire midface. In such situations, which frequently involve bilateral fracture lines, it is not possible to reconstruct the traumatized hemiface using the appearance of the unaffected half (left–right comparison method).

The N–D and Rhi–D distances are necessary for the restoration of the bony support of the medial cantal ligament and of the lacrimal apparatus. In most cases of comminuted fractures, such as type III Markowitz NOE fractures, when the bone support of the medial cantal ligament is destroyed, the accurate detection of point D makes it possible to create a new attachment of this ligament, either by trans nasal wire fixation or directly on the implantable material. From a surgical point of view, it should be considered that the supporting structures are much more complex, comprising muscles, fascia, tendons and other ligaments [26].

The Rhi–FMO and N–FMO distances are of real help in the case of lateral orbital wall fractures when the restoration of the orbital contour as accurately as possible offers superior aesthetic results. Other points, segment or angle measurements would be required for orbital reconstruction, especially for the orbital floor [27]. For results with a satisfactory accuracy, the relationship of bone structures with the periorbital soft tissues should not be ignored [28]. CT examination should be used to dynamically evaluate the immediate and subsequent effects of reconstructive surgery [29].

The N–Rhi, Rhi–IOF, N–IOF distances are used for the reconstruction of the nasal pyramid, both in cases of nasal fractures, as well as in cases of complex fractures of the facial massif or of the NOEM complex. Being located in the central area and occupying an important position on the face, the nose has one of the most important aesthetic implications. It is already known that nasal reconstruction is the oldest form of plastic surgery, being constantly improved over the centuries [30].

Performing direct measurements on human skulls is the method of choice in assessing the distances and dimensions of the viscerocranium, but it presents some difficulties both ethically and logistically. By using virtual models, the number of included subjects can increase, thus improving the accuracy of the measurements. Even with virtual 3D models it is mandatory to take into account the ethical aspects and the protection of personal data [31]. A study [32] comparing the veracity and trustworthiness of measurements on 3D models obtained from CBCT examinations with standard cephalometric measurements concluded that this type of measurement, using virtual models, can be used as a diagnostic method.

The performed anthropometric measurements bring a number of advantages in the preoperative preparation of the reconstruction process but also allow the adoption of an individualized treatment for each patient. Measurements performed on the bone were chosen for the study, considering the fact that, for a satisfactory, stable aesthetic result, the accurate reconstruction of the resistance structure must be a priority. Thus, the obtained measurements are useful in cases of complex fractures of the midface and NOEM complex, when it is not possible to compare the two hemifaces, or in cases of communitive fractures, when it is necessary to remove fractured bone fragments and restore contours and bone continuity with bone grafts, flaps or titanium implantable material (TiMesh).

Landmark based craniometry has some disadvantages described in some studies [33,34]. The authors refer to the possibility of encountering a landmark localisation error, the magnitude of which depends on the landmark category. Additionally, the method is time-consuming and can be influenced by the level of fatigue and expertise of the observer. In this study, the anthropometric points utilised were type I and type II landmarks, according to Bookstein [35], making them easier to recognize. Point positioning and measurements were checked by two other clinicians. If any doubt appeared regarding the placement of any point, the 3D model was discarded.

The three-dimensional craniometric measurements regarding midfacial dimensions and proportions is not quite an extensively explored topic, considering the fact that most of the studies already published refer mostly to the orthognathic surgery field. To our knowledge this is among the first studies that evaluate the normal distances between the chosen craniometric points on a virtual 3D model of the human skull.

## 5. Conclusions

Significant differences between genders were observed for the orbital width (N–FMO, Rhi–FMO), width of the nose bridge (N–D, Rhi–D, D–D) and for the distances evaluating the nasal projection (N–Po, Rhi–Po). No significant differences were found between males and females for the distances evaluating the length of the nose (N–Rhi). Most of the measured distances were similar to those found in other studies evaluating anthropometric facial measurements in the adult population.

The measured values may be of practical importance in the context of complex fractures of the midface and NOEM structures, assisting surgeons in restoring the lost morphology of the region, even in comminuted bilateral fractures, when intraoperative landmarks are absent.

## Figures and Tables

**Figure 1 ijerph-19-10412-f001:**
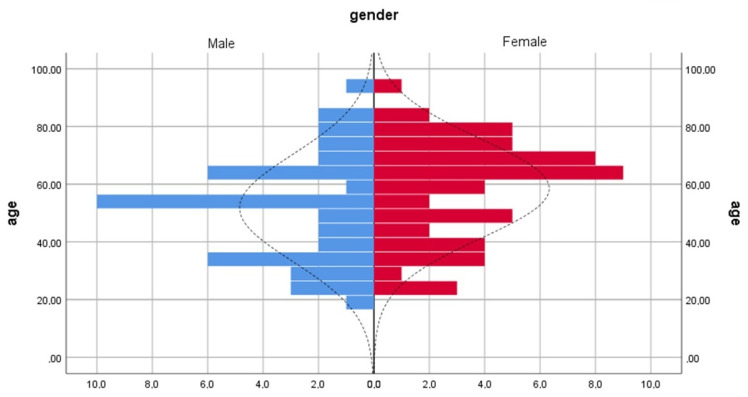
Distribution according to gender of the group.

**Figure 2 ijerph-19-10412-f002:**
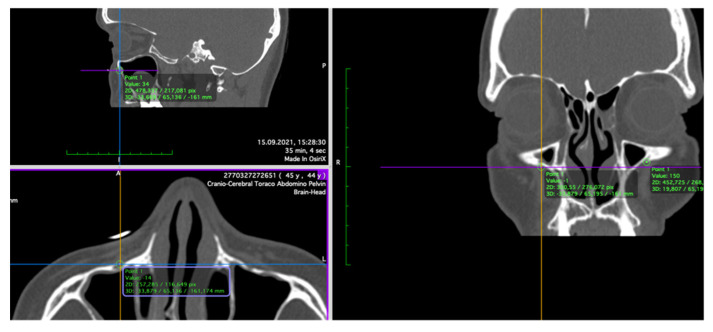
Placement of the right IOF point.

**Figure 3 ijerph-19-10412-f003:**
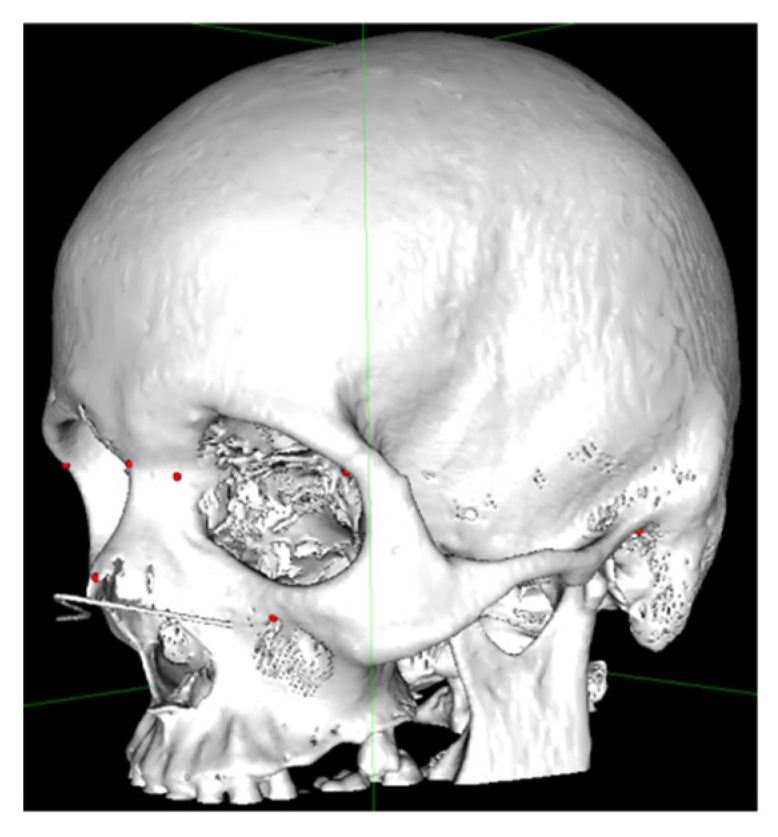
A 3D model with the marked points showed as red dots.

**Figure 4 ijerph-19-10412-f004:**
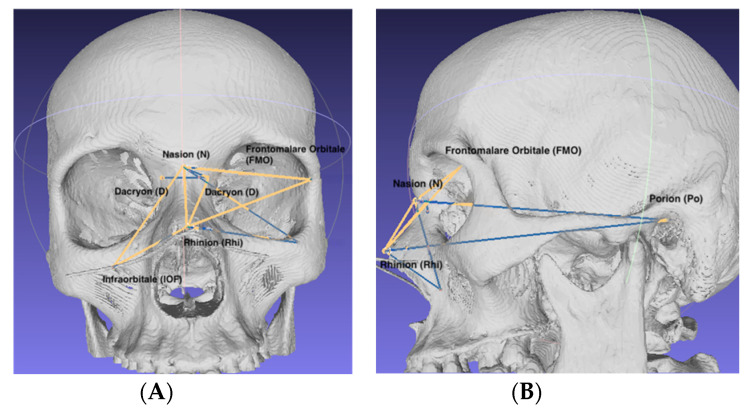
Performing measurements on the virtual 3D model; (**A**) anterior view; (**B**) lateral view.

**Figure 5 ijerph-19-10412-f005:**
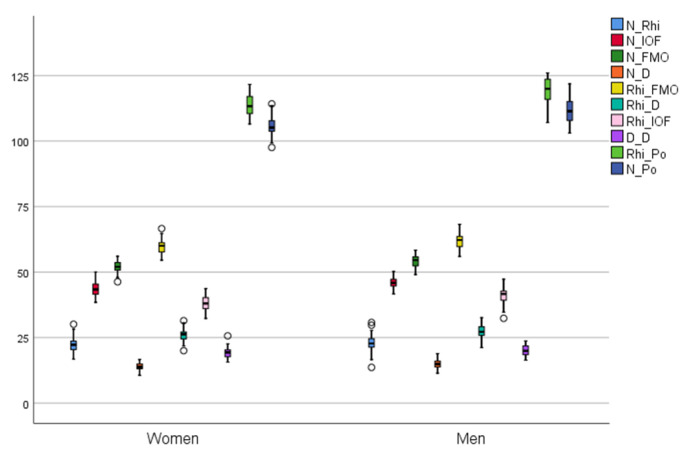
Distance values distribution according to gender.

**Table 1 ijerph-19-10412-t001:** Distance values (in mm) between the anthropometric points.

Distance	Female (*n* = 55) Mean ± SD (Min–Max)	Male (*n* = 45) Mean ± SD (Min–Max)	Total (*n* = 100) Mean ± SD (Min–Max)	*p*-Value	95% Confidence Interval of the Difference	
					Lower	Upper
N–FMO	52.07 ± 1.98 (46.28–56.00)	54.24 ± 2.19 (49.05–58.33)	53.05 ± 2.34 (46.28–58.33)	0.000	1.33	3.00
N–Rhi	22.18 ± 2.75 (16.75–30.09)	22.88 ± 3.33 (13.61–30.76)	22.50 ± 3.03 (13.61–30.76)	0.254	0.50	1.90
N–IOF	43.43 ± 2.55 (38.48–50.05)	46.10 ± 2.57 (41.65–56.01)	44.63 ± 2.87 (38.48–56.01)	0.000	1.64	3.69
N–D	13.92 ± 1.42 (10.60–16.64)	15.01 ± 1.76 (11.43–18.87)	14.41 ± 1.66 (10.60–18.87)	0.001	0.45	1.72
Rhi–FMO	59.72 ± 2.52 (54.54–66.61)	61.76 ± 2.90 (55.98–68.21	60.64 ± 2.87 (54.54–68.21)	0.000	0.96	3.11
Rhi–D	25.99 ± 2.24 (19.99–31.45)	27.21 ± 2.77 (21.19–32.57)	26.54 ± 2.55 (19.99–32.57)	0.016	0.22	2.22
Rhi–IOF	38.08 ± 2.68 (32.29–43.68)	40.93 ± 3.13 (32.34–47.24)	39.36 ± 3.21 (32.29–47.24)	0.000	1.70	4.01
D–D	19.16 ± 1.93 (15.63–25.64)	19.98 ± 1.94 (16.37–23.72)	19.53 ± 1.97 (15.63–25.54)	0.038	0.04	1.59
Rhi–Po	113.89 ± 3.87 (106.53–121.62)	119.59 ± 4.67 (107.10–125.96)	116.46 ± 5.10 (106.53–125–96)	0.000	4.00	7.39
N–Po	105.80 ± 3.31 (97.56–114.21)	111.47 ± 4.37 (103.08–121.82)	108.35 ± 4.75 (97.56–121.82)	0.000	4.14	7.20

## Data Availability

Not applicable.

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
