# Peer review of "Improving the Treatment Outcome of Naso-Orbito-Ethmoido-Maxillary Fractures Using Virtual Three-Dimensional Anthropometric Data"

_ijerph, 2022, doi:10.3390/ijerph191610412_

Round 1
Reviewer 1 Report
Dear authors,
You have conducted an interesting study with good clinical value in the field of maxillofacial surgery and traumatology. It is a descriptive study that provides useful information for the reconstruction of the midfacial structures after detrimental injuries. I have a few suggestions for improving the manuscript, which you can find below. Best regards.
Minor comments
- Title:
Please consider revising to: Improving the treatment outcome of Naso-Orbito-Ethmoido-Maxillary Fractures Using Virtual Three-Dimensional Anthropometric Data.
- Abstract:
In the “results” section, please describe briefly what the significant measurements represent. It is not possible to explain all the abbreviations in the abstract, but at least mention the meaning of the important findings.
- Materials and Methods:
There is no mention of the machine used to acquire the CT scans. Please add necessary information.
- Results:
a. Please check the following values for possible mistakes: “For the sagittal plane, the Rhi – Po distance had a mean value of 113,89±3,87 mm for the 210 female subjects and 119,59±4,67 mm for the male subjects. As for the N – Po distance, the 211 mean values were 105,80±3,31 mm and 11,47±4,37 mm respectively.”
b. Table 2 can be added as supplement. It does not add important information to the manuscript.
Major comments
- Materials and Methods:
a. There is no mention of the statistical analyses that were performed to determine differences between males and females. There are several tests and p-values mentioned in the results, but a section about “statistical analysis” is missing in the materials and methods. Please add the necessary information in this section.
b. There is a large age range in the sample. Age is a confounding factor when craniofacial form is studied, and this needs to be taken considered when conducting the statistical analysis and when interpreting the results.
Reviewer 2 Report
The authors has made a significant revision to to manuscript based on previous comments by reviewers. This edition of the manuscripts is significantly improved compared to the initial version. The clinical correlation and significance of these landmark has also be made clear. I believe the it this current form, it is acceptable for publication.
Just on additional comment, was there any interrater agreement calculation made. It seems important as there are 3 different surgeon assessing the landmarks. Thank you
Reviewer 3 Report
I see that the authors have extensively improved the paper based on previous comments. However overall I would suggest a thorough language, scientific and structural editing of the complete manuscript as in the current state it very difficult to read due to a very haphazard flow of the writing.
- Line 151: define infraorbital foramen
- Line 206: there are no groups now. You can stick with the term gender (male/female)
- At the end of methodology add statistical analysis paragraph (which software, tests for normality, comparison , significance etc)
- 223-336 Remove this part and remove Table II from the manuscript as there as there is no need because data is normally distributed. Instead just add a statement at the very beginning of results, under lines of “All the values/data were normally distributed when tested by both Kolmogorov-Smirnov and Shapiro-Wilk test”
- Line 165-168 I would suggest repetition of the experiment for atleast 10% of the cases by all observers for assessing the intra- and inter-observer reliability of all measurements through ICC to check for observers variability. Plus kindly mention experience in years of all observers.
- Add sample size calculation either in methodology before the sample or in results.
- Remove the word “group” from table 1 and everywhere in the manuscript just say “male” “female” “total”
- Romove mean+-SD min max from columns and add in rows. Upper and lower limits should have same number of digits after decimals as mean (.00)
- In significance testing only show p value in brackets (p=.000…) and remove (t(98 etc) from the complete section and abstact
- Line 201-212 reporting should be concise as it is just duplication of table 1.
- Stick with the terms “male,female” or “men, women”, do not interchange.
- The results are too limited, thereby I would suggest shortening the discussion as it is way too long and the readers might get lost in between. You should stick with the format of introductory paragraph, followed by discussion of results, clinical significance, limitations and future recommendations. then conclusion
- Line 391 reasoning for small difference? Line 396 same thing. Anywhere where there is contradiction to literature, reasoning should be provided. Or you can add a general statement under the lines of …...this could be due to selection of a different population group or anything else which might support your findings.
- Conclusions should be rephrased and remove line 689-691. The conclusion in abstract is much better. You can say under the lines of that these measurements could act as a reference guide for surgeons……
Author Response
Please see the attachment.

This manuscript is a resubmission of an earlier submission. The following is a list of the peer review reports and author responses from that submission.
Round 1
Reviewer 1 Report
The authors have presented a paper on 3D anthropometric evaluation for improving NOEM fractures outcomes.
Overall, the language of the manuscript is fine. However major flaws exist related to the reporting and linking with NOEM and the clinical connection created by the authors is very artificial not linking to the results at all.
Please see comments below,
Abstract
Complete abstract should be rewritten with Background, aim, methodology, results and conclusions. Right now it is just an extended background and aim.
Its “naso-orbito-ethmoido-maxillary (NOEM) complex” NOT “Naso-orbital-ethmoido-maxillar”. Kindly change in the whole manuscript where required.
Introduction
It should include what is known and unknown about the topic at hand which is 3D anthropometric measurements. Following that the aim should be written.
There is a lack of references. More literature search should be conducted focusing on the topic.
Materials and methods
Page 2 line 53: correct spelling “comity” to committee
Page 2 line 62: What was the data distribution of both groups based on age. How many male and female and age range. Kindly mention.
Page 2 line 71: Frontomalare Orbitale- I fail to see this landmark in the reference provided.
Page 2 line 72: kindly explain the segmentation process (conversion of DICOM to STL). How was it done. What was the threshold, was it semi-automatic. What do you mean by artefact removal, which artefacts are the authors referring to?
Was the suture visible on the 3D model clearly for placing FMO landmark?
Who performed the segmentation and evaluation of landmarks. Observer specialty and experience?
Which distances and/or planes were evaluated, this information is kind of available in results not in methodology. Kindly add and define.
The observer reliability and validation of the methodology is lacking related to both segmentation and landmarks placement.
The methodology is prone to accumulative error more if done directly on 3D models and also depending on the landmarks. The landmarks selection and placement should have been done or reconfirmed on the CBCT axial, coronal and/or sagittal planes instead of a 3D model for an accurate representation and I am not sure whether the authors did it. I agree with reference 20 provided if you also used “3D Cephalometric module”.
Statistical analysis
Proper Statistical analysis is missing and only descriptive analysis has been done. No significance testing.
Results
Too many tables added for each variable. This is gona happen if you are trying to report descriptive data rather than proper statistical testing. The authors are just trying to report all the raw data which is ok for a thesis but not a manuscript. Kindly take advice from statistician for proper reporting, checking for data normality and analysis.
Discussion:
The authors are trying to link the landmarks assessment to NOEM fracture which isn’t making a proper sense. More than half of the discussion is a clinical discussion without having any link to the reported results. The discussion should report and compare findings of the already present literature.
Page 13 line 216 introducing deep learning has nothing to do with the current aim or methodology based on landmkars. Focus should be on results and then limitations should be added at the end of the discussion with future recommendations on whether to automate the landmarking process etc. The study the authors are referring to is a classification study and the current study is not classifying fractures rather than just reporting anthropometric measurements.
The authors have wrongly concluded and written so many clinical things in the discussion just based on the evaluation of 5 landmarks.
If authors just focused on 3D anthropometry with an innovative methodology and added NOEM and other facial reconstructive features as a clinical relevance that would make more sense.
The authors also have to realize that we are moving away from landmarks based evaluation methods due to a high degree of variability and human error.
Reviewer 2 Report
The authors performed a Three-Dimensional Anthropometric measurement on a specific population in Romania. Generally the methods are sound but some concerns relating to the usefulness of these measurements clinically due the omission of antroposterior measurement. The discussion section must also be improved. All these concerns are detailed below:
1. It is not clear or explained why the group were divided and why 60 years old was the cutoff age
2. Repair of NOE structures need measurement in axial, coronal and sagittal plane. Most of the measurement here are coronal and axial plane but no sagittal plane, which is important in the NOE prominence antero-posteriorly. Especially the depth of the deepest point between nasion and rhinion
3. The figure 1 should also include image from the lateral view to show the landmark position relationship in sagittal plan
4. The authors state that these measurements may be use as a guide in the NOE fracture repair. However they do not give detail on the significance of each measurement in clinical practice. I would suggest the authors provide some example on how one specific measurement can be use in clinical practice. To add in discussion
5. The authors also fail to state population on which the applicability of these measurements could be used. Is it just for Eastern and North Eastern region of Romania population? How do these measurements compares with previous measurement in different population? To add in discussion
Round 2
Reviewer 1 Report
The revision conducted by the authors is appreciated. Response to some comments is unsatisfactory. At the same instance, the methodology still lacks novelty and conclusions are far fetched.
Reviewer 2 Report
I appreciate the authors efforts to try the very best to address reviewers comments. However, I do feel the revision did not alleviate the initial concerns enough. The applicability of such measurements in clinical situation is still not clearly explained. While the authors explained the importance of the specific measurement/distance in NOE fractures, but how can it be use it clinical practice? How do we transfer these measurement in operating theater? I which population can this measurement be used specifically? Broad statement as added in discussion is not sufficient to bring forward clinical applicability of such measurement especially in this day and age where real time guided surgery is prevalent. The authors also gave explanation on the age cutoff (60 years old) being osteoporosis, but how does osteoporosis relate to NOE fractures is not clear?